

# Beyond 2D cell cultures: how 3D models are changing the *in vitro* study of ovarian cancer and how to make the most of them

Marilisa Cortesi[1,2], Kristina Warton[1] and Caroline E. Ford[1]

[1] School of Clinical Medicine, University of New South Wales, Sydney, New South Wales, Australia
[2] Department of Electrical Electronic and Information Engineering "G. Marconi", University of Bologna, Cesena, Italy

## ABSTRACT

3D cell cultures are a fundamental tool in ovarian cancer research that can enable more effective study of the main features of this lethal disease, including the high rates of recurrence and chemoresistance. A clearer, more comprehensive understanding of the biological underpinnings of these phenomena could aid the development of more effective treatments thus improving patient outcomes. Selecting the most appropriate model to investigate the different aspects of cell biology that are relevant to cancer is challenging, especially since the assays available for the study of 3D cultures are not fully established yet. To maximise the usefulness of 3D cell cultures of ovarian cancer, we undertook an in-depth review of the currently available models, taking into consideration the strengths and limitations of each approach and of the assay techniques used to evaluate the results. This integrated analysis provides insight into which model-assay pair is best suited to study different parameters of ovarian cancer biology such as cell proliferation, gene expression or treatment response. We also describe how the combined use of multiple models is likely to be the most effective strategy for the *in vitro* characterisation of complex behaviours.

## INTRODUCTION

Ovarian cancer delineates a set of malignancies sharing anatomical location and characterised by late-stage diagnosis, high rate of recurrence and rapid development of treatment resistance (*Siegel, Miller & Jemal, 2020*). Prognosis for this disease is generally poor with a 5-year survival rate below 50% and treatment is still largely reliant on chemotherapy using platinum-based agents combined with paclitaxel, and debulking surgery (*Gaitskell et al., 2022*). This approach often has good efficacy at first diagnosis but is not generally beneficial at relapse, which occurs in about 80% of patients (*Garzon et al., 2020*). Indeed, surgery at relapse might not be indicated and resistance to chemotherapy is common. Some targeted treatments are available, notably PARP inhibitors (*Xie et al., 2022*), but the limited number of established markers of disease, together with high

Corresponding authors
Marilisa Cortesi,
marilisa.cortesi2@unibo.it
Caroline E. Ford,
caroline.ford@unsw.edu.au

variability and diversity of ovarian cancers complicate the development of new effective treatment options.

Achieving this objective relies on appropriate pre-clinical models to study ovarian cancer biology and test treatment efficacy. However, this cancer is characterised by features poorly recapitulated by 2D monolayers, such as a strong interaction with the extracellular environment and the stromal and immune cells therein. In recent years, 3D cell culture models of ovarian cancer have become more widespread as they enable the *in vitro* study of cell–cell and cell–matrix interactions and replicate the microenvironment variability characteristic of *in vivo* growth (*Jensen & Teng, 2020*). A key feature in this regard is their ability to recapitulate the oxygen gradient and hypoxic core often observed within solid tumours (*Godet et al., 2022*). These harsh environmental conditions have been repeatedly shown to promote cancer cell aggressiveness and drive drug resistance (*Godet et al., 2022*). This is particularly relevant in ovarian cancer where reduced access to oxygen has been shown to hinder treatment response in patients (*Klemba et al., 2020*). 3D cultures accurately model this phenomenon thus supporting its effective characterisation in a controlled environment.

The ability to better approximate *in vivo* behaviour holds the potential of improving the accuracy of *in vitro* drug testing. Indeed, the limited *in vivo* transferability of laboratory results is a major contributor to the low success rate of the current drug development pipeline. Better *in vitro* models would also contribute to the implementation of the 3R strategy, which aims at reducing and replacing the use of animal models (*Melzer et al., 2019*).

These 3D models are—by necessity—a simplification. Differences in composition and structure with respect to the system they aim to replicate result in discrepancies in behaviour and the inability to capture all features observed *in vivo*. However, rather than being a limitation, this reduced complexity is a key feature, as fewer elements and interactions aid the interpretation of results and enable a higher degree of control on the experimental setting. A more careful consideration of the model itself, however, becomes necessary as each assumption and simplification might affect the results, thus potentially leading to inconsistencies between different models of the same biological process.

In this review, all the major ovarian cancer 3D cell culture models are compared within the context of the quantification of specific cell properties that can be measured in the model. This focus on appropriate matching of 3D model and assay sets this analysis apart from recent articles published on similar topics (*Ciucci et al., 2022*; *Watters, Bajwa & Kenny, 2018*; *Yee et al., 2022*; *Yang, Xu & Zhao, 2020*; *Braccini et al., 2022*; *Tsang et al., 2022*; *Liu et al., 2020*; *Nero et al., 2021*) and provides a methodological framework for the evaluation of the features and simplifications of the models whose validity is general and extends beyond the currently available 3D cultures. As such, other researchers interested in using 3D culture models are the main intended readership.

## SURVEY METHODOLOGY

References were retrieved on NCBI PubMed and Google Scholar. The keyword "ovarian cancer" in conjunction with either "3D *in vitro* models", "3D culture models" or the

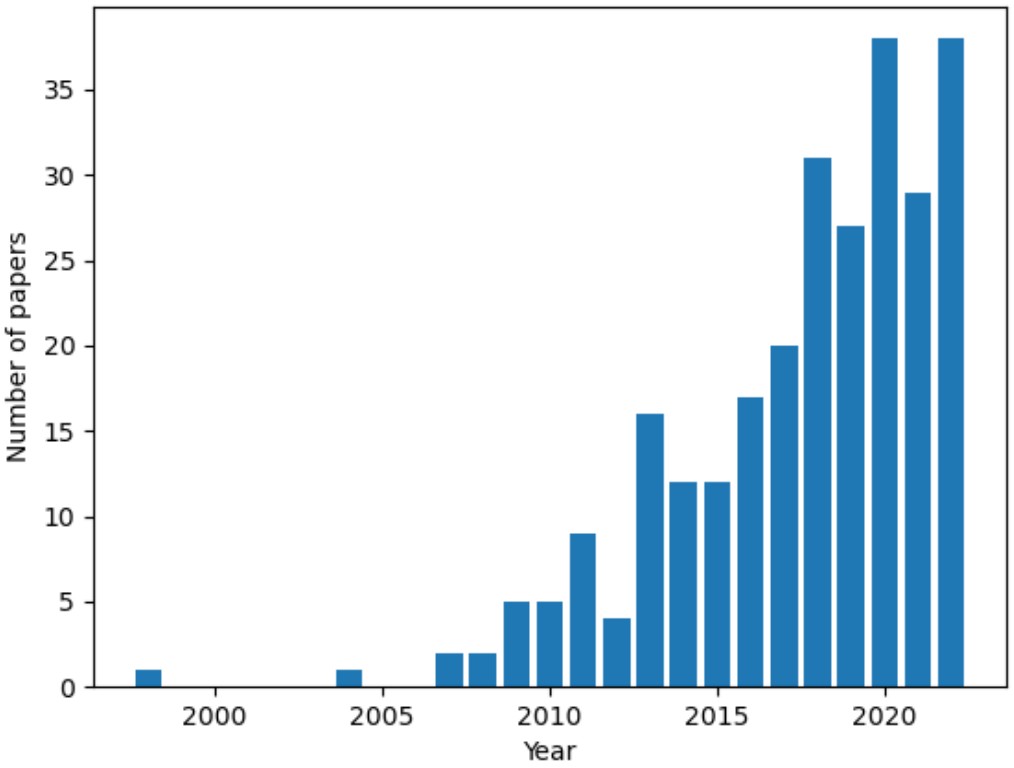

**Figure 1** **Number of articles over time for the query "3D culture models ovarian cancer".** Data retrieved from NCBI PubMed. Figure created using custom code in Python v3.9.

name of a specific culture setup (spheroid, scaffold, organotypic model and organoid) was used in the search. A growing interest in the topic is demonstrated by the steep rise in the number of published articles over time (Fig. 1). The first publication returned by the query "3D culture models ovarian cancer" is from 1998 and there is a 10-fold increase in the number of published articles between 2012 to 2022. This review focused on studies from the 5-years preceeding the survey (September 2022), to provide a current and relevant overview of the topic. Key works from previous years were also included to improve the comprehensiveness of our analysis.

### *In vitro* models of ovarian cancer

To improve the clarity of the next sections and provide an overview of the terminology used throughout this work, we here briefly describe the major classes of *in vitro* models of ovarian cancer. References to specific models and protocols will be provided in the following sections.

Figure 2 provides a summary of these systems sorted by complexity. The simplest set-up (model A) consists of cells cultured on 2D on polystyrene surfaces (*e.g.*, petri dishes, flasks). This is an established approach, and a wealth of resources are available for set-up and analysis of standard *in vitro* assays. It however represents an extreme simplification of

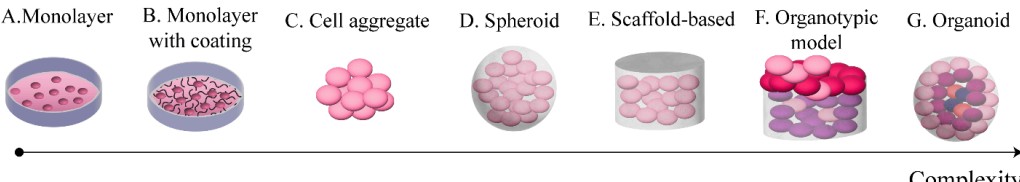

Complexity

**Figure 2  Commonly used *in vitro* models of OC.** These models are sorted from least (A) to most (G) complex.

the *in vivo* environment, limiting the extracellular matrix (ECM) and constraining cells on a bidimensional surface.

Protein coatings (model B) represent a slight increase in complexity. They involve covering the polystyrene surface with a protein substrate (*e.g.*, collagen or fibronectin) prior to seeding the cells. This model allows for a simplified cells-ECM interaction, without losing the ease of use of the monolayer model. It is however worth noting the 2D nature of the interaction between the cells and the substrate, which, depending on the application, might not be sufficiently accurate.

Cell aggregates (model C) are the simplest 3D culture model. They consist of clusters of cells adhering to each other without any external support structure. As such, they are generally small and of approximately spherical shape. Some of the assays developed for 2D cultures can be applied to these models with minimal modifications, as their size and looseness do not generally interfere with the functioning of the assay. These features make cell aggregates potentially effective for the study of cell–cell interaction and cell auto-organisation in the 3D space. Cluster compactness has however been shown to depend on the cell type (*Al Ameri et al., 2019*), thus limiting this model to specific cell lines and short-term experiments.

The next level of complexity involves the addition of an external support matrix, which serves the double purpose of improving the structural stability of the culture and introducing an ECM surrogate. Both spheroids (model D) and scaffold-based models (model E) belong at this complexity level, but several differences distinguish them. Spheroids are cells-material aggregates of approximately spherical shape; they are relatively small and are produced with the cells embedded in the material, typically by resuspending the cell population in a solution of culture media and surrogate ECM and inducing gelation through temperature changes or other methods that preserve cell viability. While the term spheroid is often used to identify spherical cell clusters irrespective of the presence of an exogenous support material, we here distinguish between cell aggregates and spheroids as the assumptions and simplifications of the two models are different.

In contrast to spheroids, scaffold-based cultures generally require cells to be seeded subsequently to scaffold production and rely on moulds to give shape to the culture (*Paradiso et al., 2019*). These set-ups can vary greatly in shape, size and ECM composition, thus potentially enabling a wider range of models to be developed and a greater flexibility in their features.

Spheroids and scaffold-based set-ups enable the study of 3D interaction both among cells and with the extracellular environment. In some examples, the co-culture of multiple cell types adds a level of complexity and allows for the study of interactions between cancer cells and healthy tissue (*Hedegaard et al., 2020*; *Tofani et al., 2021*). The higher density and increased size of these cultures additionally result in different microenvironments which give rise to differential behaviours within a single structure.

The most complex set-ups currently available, organotypic models (model F) and organoids (model G), aim at replicating structure and functions of entire organs through the combination of external support materials, multiple cell types and structural and organisational features typical of the reference tissue (*Youhanna et al., 2022*). The main difference between them is that organotypic models are constructed layer by layer and are thus characterised by a higher reproducibility and control over their composition, while organoids self-assemble combining the cells and ECM available within primary tissue samples. This increases their variability and makes the success rate of their formation dependent on the characteristics of the sample itself (*e.g.*, subtype, tumour cell content) (*Kopper et al., 2019*; *Nanki et al., 2020*).

While 3D models recapitulate tissue structure and cell–cell interactions more closely than 2D models, their laboratory analysis is a critical limitation. Most 2D native protocols might not be applicable, especially to large dense cultures like organoids or organotypic models. While assays specific to 3D settings are being developed (*Cortesi & Giordano, 2022*), 3D constructs are currently mostly handled in the same way as tissues, with fixing, cutting and staining being standard *de-facto*. These requirements preclude the possibility of longitudinal measurements and increase the variability in the results, as differences in initial conditions (*e.g.*, cell number and distribution, material composition) cannot be accounted for. Because of this variability, a larger number of cultures will be required for each experiment, thus increasing the resources necessary. It is important to note that this consideration is less critical for simpler culture models (*e.g.*, spheroids, cell aggregates) that benefit from the availability of accurate reliable protocols for their construction (*e.g.*, 96 well round bottom plate) and a wider array of non-destructive *in vitro* assays.

Organotypic models and organoids are further complicated by the necessity of distinguishing the contribution of the different types of cells. They, however, hold the most potential as they are the most accurate recapitulation of *in vivo* structure. In addition, these models generally require the use of primary cells, thus enabling by default the study of individual variability.

Several *in vivo* models of ovarian cancer have also been developed (*Tudrej et al., 2019*). While their comprehensive analysis is beyond the scope of this review (this topic is discussed in detail in *Tsang et al. (2022)*; *Tudrej et al. (2019)*; *Kim et al. (2020)*), it is worth noting that they also exhibit a wide range of features and varying complexity. Fruit flies are the simplest model and are generally used to study specific cell behaviours (*e.g.*, invasion) and gene expression changes. At the other end of the spectrum laying hen and patient derived xenograft mouse models are used to study the spontaneous development of ovarian cancer and patient variability, respectively. As with *in vitro* cultures, each *in vivo* model has specific strengths and limitations that need to be taken into account. A major difference however is

the reduced control of the experimental setting that animal models afford (*e.g.*, interaction among different systems/organs which vary among species and often individuals). While this is often cited as a key feature of animal models, that makes them necessary to test the safety of new treatments, it is worth noting that multiple evidences point at these interactions not being conserved across species (*Akhtar, 2015*) thus leading to potentially misleading results, which would be difficult to identify due to the incomplete knowledge regarding these changes and how they relate to human biology.

*In vitro* models are better suited to study cell behaviour, and the complexity achieved by integrating multiple cell populations and complex tissue-like organisation could provide valuable and relevant insights. For example, systematically comparing different 3D models enables the study of the role of individual factors in cell behaviour. Differences in the results obtained in cell aggregates and spheroids using equivalent experimental conditions could shed light on the contribution of the ECM, while a comparison of co-cultures conducted in spheroids, scaffolds and in organotypic models could help understand how the structural organisation of the tissue influences the experiment's results.

Finally, it is worth noting that the complexity classification presented in this section is just a useful tool to summarise the wide range of models available for the *in vitro* study of OC. As such, the reader is invited to consider the complexity spectrum as a continuum and the model classes as reference points throughout the analysis that follows.

## Cell growth and proliferation

The quantification of cell growth and proliferation is almost universal in *in vitro* studies, as it is a proxy for cell health. It is also a key measure of treatment response and a major indicator in biocompatibility studies. In this section we will focus on cell growth and proliferation in the absence of pharmacological treatment, as the evaluation of drug response will be the focus of a subsequent section.

All the major classes of 3D monoculture models are used for the quantification of cell growth, in conjunction with a wide variety of assays (Table 1). Metabolic and staining-based methods are the most common, but DNA-based approaches are also available.

### Metabolic assays

Metabolic assays rely on quantifying the concentration of a byproduct of cell metabolism to provide an estimate of the amount of living cells in the culture. Examples of metabolic assays include Alamar Blue, MTT and CCK8 assays. While originally developed for 2D cultures, these assays are extensively used in 3D settings (Table 1), together with commercial kits specifically developed for 3D models (*e.g.*, CellTiter Glo 3D).

Overall, these assays show delayed growth and a reduction in proliferation when cells are maintained in a 3D environment compared to 2D (*Al Ameri et al., 2019*; *Tofani et al., 2020*; *Liu et al., 2018*). This is partly compensated by the ability of 3D settings to sustain larger cell populations for longer time periods (*Liu et al., 2018*). Another interesting result is presented by *Hedegaard et al. (2020)*, where cell growth was shown to be dependent on the surrogate ECM. While the qualitative growth dynamic and lag phases were fairly consistent, a significant variation in cell number between different matrices was observed, pointing to the need for the careful selection of the 3D culture's support structure.

**Table 1** Summary of the main 3D models and assays used to evaluate cell growth and proliferation.

| 3D cell culture model | Assay | Citations |
|---|---|---|
| Cell aggregate | Aggregate size | *Singh et al. (2021)*, *Hirst et al. (2018)* |
| | Ki67 staining | *Hirst et al. (2018)* |
| | Live-dead assay | *Hedemann N et al. (2021)*, *Brodeur et al. (2021)* |
| | Metabolic assay (MTS, CCK8) | *AlAmeri et al. (2019)* |
| Spheroids | BrdU incorporation | *Song et al. (2020)* |
| | DNA quantification | *Song et al. (2020)* |
| | Ki67 staining | *Pietilä et al. (2021)* |
| | Live-dead assay | *Hedegaard et al. (2020)*, *Song et al. (2020)* |
| | Metabolic assay (CellTiter-Glo, Alamar blue, MTT) | *Hedegaard et al. (2020)*, *Tofani et al. (2020)* |
| Scaffold based | DNA quantification | *Paradiso et al. (2019)* |
| | Live-dead assay | *Liu et al. (2018)* |
| | Metabolic assay (CCK8) | *Liu et al. (2018)* |
| Organoids | Size and number | *Kopper et al. (2019)*, *Nanki et al. (2020)*, *Maenhoudt et al. (2020)*, *Hoffmann et al. (2020)* |

An important limitation of metabolic assays is their dependence on diffusion of the reagent through the culture. This can make them unsuitable for larger, denser models where concentration gradients within the structure will create differential access to the reacting solution thus weakening the correlation between the reaction product and the number of living cells. It is also important to note that these methods become less reliable when cell metabolism changes between the tested conditions, such as when studying radiation-induced growth inhibition (*Rai et al., 2018*).

Despite these caveats, metabolic assays are simple to use and highly standardized, with dedicated commercial kits available. As such, they represent a viable option, especially for smaller, looser culture models such as cell aggregates and spheroids.

### DNA-based assays

DNA quantification and BrdU incorporation provide population-level measurements of cell density and proliferation. They rely, respectively, on the measurement of the total amount of DNA in the sample and on the quantification of the fraction of BrdU, a thymidine analogue, incorporated in newly synthesised DNA. Two articles using these methods show almost linear growth curves throughout the considered time frames (6- or 9-days post seeding) (*Paradiso et al., 2019*; *Song et al., 2020*). Interestingly, results obtained with DNA-based assays seem to be associated with a reduced growth delay when compared with metabolic assays. While, to the best of the authors' knowledge, no direct comparison between these two approaches has been published, it suggests that DNA quantification might be more accurate than metabolic methods. Indeed, the finite diffusion velocity of the reagent and/or a transient slowdown in cell metabolism seem to be important confounding factors, especially at the earlier timepoints. The high frequency of aneuploidy in ovarian cancer, however, is a potential drawback of this technique, as the amount of DNA present

within each cell is difficult to estimate *a priori* and varies with disease stage (*Friedlander et al., 1983*). DNA-based assays, however, represent a valid alternative when the cells ploidy is known or constant (*e.g.*, monitoring of the same culture over time).

### Staining assays

Staining-based methods mainly consist of live-dead assay and Ki67 quantification. The former uses two different dyes, which selectively accumulate in living or dead cells, while the latter quantifies the expression of Ki67, a protein with key roles in the cell cycle, whose intensity and localization are markers of proliferation (*Sun & Kaufman, 2018*). These techniques are mostly used as qualitative evaluations of cell health, even though image analysis pipelines for quantification of the signals are available (*Lovecchio et al., 2019*; *Pasini et al., 2021*; *Feng et al., 2020*) and flow cytometry can be applied if cell retrieval from the culture is possible (*Brodeur et al., 2021*).

In most cases, the staining procedure is applied to the entire culture, *Hirst et al. (2018)*, however, stained the cross section of compact spheroids, to highlight the presence of a hypoxic core. Proliferating cells were mainly concentrated toward the edge of the structure, while the center of the scaffold exhibited a stronger stain for pimonidazole, a commonly used hypoxia marker (*Ragnum et al., 2015*). Coupling the study of cell viability and distribution within the culture with the analysis of hypoxia and/or apoptosis is an effective strategy that yields a comprehensive picture of cell status and behaviour.

While culture compactness and resource availability are key drivers in the formation of a hypoxic core, which might not be present or as prevalent in other systems, the increased spatial resolution afforded by staining methods is a key feature that enables a deeper understanding of 3D cell culture behaviour. Appropriate sampling of all the different microenvironments within the model, however, becomes fundamental to avoid biases in the results.

Overall, staining-based methods enable the evaluation of cell growth and proliferation with high resolution and, as such, could provide unique and useful insights. Improvements in the quantification of the results and in protocol standardisation are, however, necessary.

## Culture size

Spheroid, organoid or cell aggregate diameter, measured from either static microscopy images (*Hirst et al., 2018*; *Hoffmann et al., 2020*) or from videos (*Singh et al., 2021*), is often considered as a proxy of cell number. While this approach is the norm for the study of organoid growth and has been used to characterise patient-specific growth patterns (*Maenhoudt et al., 2020*), its use in cell aggregates and spheroids has some limitations. Indeed, different cell lines have been shown to yield spheroids or aggregates with different levels of compactness (*Al Ameri et al., 2019*; *Hirst et al., 2018*; *Tofani et al., 2020*), thus weakening the relationship between size and number of cells. *Hirst et al. (2018)*, additionally, showed that there is no correlation between spheroid diameter and number of proliferating cells further undermining the role of culture diameter is as an indicator of cell growth.

### Overall analysis

A reduction in cell proliferation, when compared to 2D settings, is consistently seen in 3D cultures, irrespective of model and assay. Therefore, longer experiments might be needed. This necessity makes long term structural stability of the cultures an important requirement. While few authors commented on this aspect, *Hedegaard et al. (2020)* observed a decreasing stability of Matrigel constructs over time, with only about 60% reaching the end of a 21-day experiment. This problem seems to be less relevant for organoids, with some authors reporting successful cultures lasting several months (*Maenhoudt et al., 2020*). The improved stability of organoids might be connected to the presence of multiple cell types. For example, *Hedegaard et al. (2020)* also showed that co-culturing cancer and healthy cells had a stabilising effect on the Matrigel cultures and could be associated with a higher proliferation rate.

This result underscores the importance of the interaction between cancer and healthy cells at the metastasis site, including the presence of paracrine signalling molecules and other environmental factors. Indeed, *Hoffmann et al. (2020)* showed that supplementation with epidermal growth factor was necessary for organoid proliferation, while medium containing the Wnt signaling agonist Wnt3a stunted growth completely. Similarly, *Velletri et al. (2022)* found that pro-aggregation factors present in the ascitic fluid were required for cell aggregate formation and cancer associated fibroblasts have been shown to increase the aggressiveness of cancer cells (*Sahai et al., 2020*). A more extensive evaluation of viability and proliferation of ovarian cancer cells co-cultured with healthy cells is thus warranted. The major obstacle to its realization is the difficulty in distinguishing between the different cell types included in the co-culture. Imaging-based methods, coupled with *ad-hoc* analysis software could be a solution to this problem, enabling the staining of markers specific for alternative cell populations and/or the evaluation of cell size and shape (*Pasini et al., 2021*; *Joshi et al., 2021a*; *Cortesi et al., 2024*).

Overall, for the quantification of cell growth and proliferation, assay selection might be more critical than model selection. Among the population-level assays, DNA-based methods may be the most accurate, as they are less dependent on culture- or cell-specific features. A higher spatial resolution is however required to observe intra-culture variability and only staining-based methods currently enable the study of how viability and proliferation vary within the culture. The laboriousness of these methods, coupled with difficulties in the accurate quantification of the results, are important limitations to their use.

## MORPHOLOGY AND DISTRIBUTION

Cell morphology and distribution are key features of ovarian cancer tumorigenesis, the *in vitro* study of which has largely been enabled by 3D cultures. Spheroids are currently the preferred model for evaluating these properties, as they allow the study of the interaction between the cells and their environment, while being simple to construct and maintain in culture. Their relatively small size, additionally, make them fairly transparent, a feature that simplifies direct visualisation without having to cut the sample. Other models, are however used, including scaffolds and organoids (Table 2).

**Table 2  Summary of the 3D cell culture models and assays used to evaluate cell distribution and morphology.**

| 3D cell culture model | Assay | Citations |
|---|---|---|
| Cell aggregates | Live dead + Hoechst staining | *Hedemann N et al. (2021)* |
| | PKH67 dye | *Velletri et al. (2022)* |
| Spheroid | Brightfield images | *Tofani et al. (2021)*, *Song et al. (2020)*, *Pietilä et al. (2021)*, *Tofani et al. (2020)*, *Liu et al. (2018)*, *Velletri et al. (2022)*, *Heredia-Soto et al. (2018)*, *Nowacka et al. (2022)* |
| | Confocal laser scanning microscopy | *Tofani et al. (2021)* |
| | DAPI/Hoechst and f-actin Network/collagen stain | *Hedegaard et al. (2020)*, *Tofani et al. (2021)*, *Hedemann et al. (2021)*, *Song et al. (2020)*, *Pietilä et al. (2021)* |
| | H&E staining | *Nowacka et al. (2022)* |
| | Scanning electron microscope | *Hedegaard et al. (2020)*, *Tofani et al. (2020)* |
| Scaffold | Brightfield images | *Liu et al. (2018)* |
| | H&E staining | *Paradiso et al. (2019)*, *Liu et al. (2018)* |
| | Scanning electron microscope | *Liu et al. (2018)* |
| | Whole mount staining | *Liu et al. (2018)* |
| Organoids | Staining and immunofluorescence (H&E, Ki67, specific proteins) | *Kopper et al. (2019)*, *Nanki et al. (2020)*, *Maenhoudt et al. (2020)*, *Hoffmann et al. (2020)*, *Lõhmussaar et al. (2020)* |

Whole culture brightfield images provide information on the overall shape, size, density and regularity of the culture, while scanning electron microscope analyses enable the high-resolution study of cell shape and of the interaction between different cells and the surrogate ECM. Staining and immunofluorescence protocols are characterised by an intermediate level of complexity and are largely used to compare organoids to the tissue of origin and to test the effect of different surrogate matrices, mostly through the visualisation of cell nuclei (DAPI or Hoechst staining) and ECM proteins, or through H&E staining.

A key drawback of these methods is the difficulty in the quantification of the results, which requires *ad-hoc* software to identify the salient components of the images (cells and matrix fibres) and conduct the subsequent evaluations. As such, in most cases only qualitative analyses are conducted. This intrinsically limits the comparison between different studies, but overall, cells cultured in 3D seem to aggregate more tightly than in 2D (*Liu et al., 2018*; *Heredia-Soto et al., 2018*; *Nowacka et al., 2021*) even though different cell lines yield varying levels of density (*Paradiso et al., 2019*; *Hedemann et al., 2021*). The surrogate ECM, its composition and concentration, are also important factors in determining cell morphology and distribution within the culture (*Song et al., 2020*; *Pietilä et al., 2021*; *Liu et al., 2018*). Morphology and distribution are often evaluated only at the whole culture level, with shape regularity and roundness being the most studied parameters (*Song et al., 2020*; *Tofani et al., 2020*).

Co-culture of cancer and healthy cells has been explored by some authors (*Hedegaard et al., 2020*; *Tofani et al., 2021*). The integration of different cell types appears to yield a more stable and consistent culture, but further investigation is warranted as in both cases it was not possible to distinguish between cancer and non-cancer cells.

Organoids generally present the same morphological and behavioural features as the tissue of origin, including histological appearance, mutation profile and treatment response (*Kopper et al., 2019*; *Nanki et al., 2020*; *Hoffmann et al., 2020*), suggesting their ability to closely mimic *in vivo* behaviour and variability between different patients. The introduction of common ovarian cancer genetic modifications, such as Trp53, Brca1, Nf1 and Pten, in murine organoid models, additionally, has been shown to result in structural and morphological changes consistent with the ones observed in patient samples harbouring the same mutations (*Hoffmann et al., 2020*; *Lõhmussaar et al., 2020*; *Zhang et al., 2021*), thus potentially enabling the replication of patient specific patterns and facilitating the study of rare mutations.

Overall, the *in vitro* analysis of cell morphology and distribution within cultures is still at the early stages of development. Progress in technical assay development and data analysis is required to enable a thorough quantification of these properties. Their importance is however indisputable and is one of the key advantages of 3D cell cultures. Among the available models, larger structures that include a surrogate ECM, and ideally multiple cell types, are the most suitable for the study of cell morphology and distribution, as they are the most likely to yield *in vivo*-like results. Their set-up and analysis, however, are more challenging than simpler systems. As such, cell aggregates and smaller spheroids are still largely used.

## CELL ADHESION

The ability to attach to a substrate is a key feature of ovarian cancer cells, as it relates to their potential for colonising healthy tissue. The experimental evaluation of cell adhesion generally relies on quantification of the number of cells that have attached to a specific substrate within a predefined time interval (*Lu et al., 2019*; *Peters et al., 2015*; *Kenny et al., 2015*; *Ritch et al., 2022*; *Cortesi et al., 2023*). While characterised by important drawbacks, such as lack of standardisation and inability of distinguishing different cell adhesion phases, this assay is very versatile and can be easily adapted to include different substrates and multiple cell types.

*Ford et al. (2020)* exploited this feature by directly comparing the ability of ovarian cancer cells to adhere to a collagen coating or to an organotypic model of the omentum (*Kenny et al., 2015*; *Ritch et al., 2022*). The equivalent patterns of adhesion retrieved with this study highlight how the increased complexity of 3D models might not be required for processes that are mostly 2D in nature. Another analysis conducted on the same organotypic model by *Cortesi et al. (2023)* used late-stage ovarian cancer cell aggregates, rather than individual cells, to evaluate adhesive properties. Interestingly, conditioned media from a confluent culture of the same cell line was required for aggregate adhesion and mesothelial layer disruption. This result further underscores the importance of paracrine signalling molecules and growth factors for the recapitulation of *in vivo* biology.

Staining of specific proteins and quantification of key markers in gene expression studies are also techniques widely used to study cell adhesion (*Paradiso et al., 2019*; *Song et al., 2020*). Indeed, they combine the quantification of the molecular determinants of behaviour with the visualisation of key structures, such as cell–cell junctions and protein localisation within the cells. When compared with 2D culture, 3D set-ups tend to be associated with a higher expression of adhesion markers (Integrin beta 1, E cad and N cad) suggesting the creation of a tighter network of cells (*Song et al., 2020*). The choice of the surrogate ECM and exogenous stimuli such as TGF $\beta$ have also been shown to influence cell adhesion, both in terms of overall affinity and temporal dynamics of this process (*Al Ameri et al., 2019*).

The evaluation of cell adhesion in 3D cultures is a well-developed field, due to the possibility of directly applying the techniques available for 2D settings. The added value of a more complex model is mostly connected with the capacity to study the interactions among different cell types and the *in vivo-* like features of the substrate. However, since many of the key steps of cell adhesion depend on surface interactions, the gap between the results measured in 2D and 3D is reduced, thus potentially favouring simpler, more manageable approaches for the evaluation of this property (*Ford et al., 2020*).

## MIGRATION AND INVASION

Ovarian cancer is particularly suited to the study of migration and invasion in cell culture, as it mainly disseminates through the transcoelomic route. This route consists of cancer cells detaching from the ovarian tumour and colonising healthy peritoneal tissues, often aided by the accumulation of liquid (*i.e.,* ascites) within the abdominal cavity (*Ford et al., 2020*). There are several approaches for the *in vitro* study of this process (Fig. 3).

Scratch wound healing quantifies migration rate by determining the time required by cells to fill a gap of predefined width (*Cortesi et al., 2017*) and can be considered a measure of the cells' ability to move on a 2D surface (Fig. 3A). A higher motility is likely associated with a more successful colonisation *in vivo*, as cancer cells can seek out regions more suitable for metastasis formation. The main drawback of this approach is that it requires cells to be seeded as a monolayer and as such it is largely a 2D technique. Interestingly, *Liu et al. (2018)* showed that pre-culturing cells in 3D induced a long-lasting increase in migration, when compared with cells maintained exclusively in 2D, which could be measured using the scratch wound healing assay. This result, besides providing a potential strategy to improve the accuracy of data measured in 2D, is coherent with evidence suggesting that cell–cell interaction in 3D is necessary to initiate metastasis in ovarian cancer (*Capellero et al., 2022*).

Another, slightly more complex approach to study cell invasion, is the transwell assay. This is the most common method for the evaluation of migration and invasion in ovarian cancer (*Al Ameri et al., 2019*; *Ritch et al., 2022*; *Coelho et al., 2018*; *Hallas-Potts, Dawson & Herrington, 2019*; *Cortesi et al., 2018*). It consists of counting the number of cells able to cross a porous membrane overlaid with Matrigel in response to a nutrient gradient. As shown in Fig. 3B this assay closely mimics the initial metastasis phases, where cancer cells

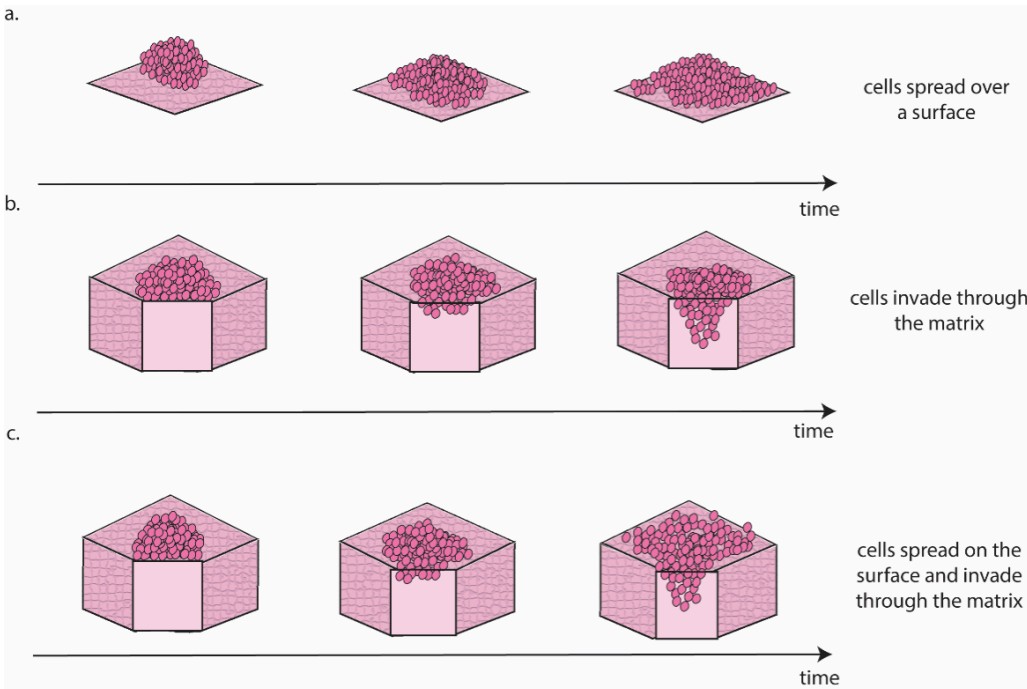

**Figure 3** **Schematic representation of the different migration and invasion modalities that recapitulate these phenomena in the lab.** (A). Migration of cancer cells on a 2D surface. This process is modelled *in vitro* using the scratch wound healing assay. (B) Invasion through a 3D matrix. This process is modelled *in vitro* using transwell invasion assays. (C) Migration on coated surfaces /invasion through gel layers combine (A and B) in a single assay. Figure created by the authors using Adobe Illustrator v27.

invade through healthy tissue. While this is generally simulated with a Matrigel layer, more accurate models that incorporate fibroblasts embedded in a collagen matrix overlayed with a layer of mesothelial cells in a structure similar to the omentum have also been presented (*Kenny et al., 2015*). This improved model is extensively used to test the response to treatment of metastatic sites (*Lu et al., 2019*; *Peters et al., 2015*) and has been shown to increase migration and invasion compared to Matrigel-based transwell assays (*Cortesi et al., 2023*).

A third approach for the quantification of cell migration and invasion in 3D consists of seeding a population of cancer cells on top of coated surfaces/gel layers and evaluating their spread and invasion over time (*Al Ameri et al., 2019*; *Paradiso et al., 2019*; *Coelho et al., 2018*; *Hallas-Potts, Dawson & Herrington, 2019*; *Borghese et al., 2020*). Like the transwell assay, this model can also incorporate different cell types, with gel layer containing healthy cells that are commonly found at the metastasis site (*e.g.*, mesothelial (*Coelho et al., 2018*) or fibroblast (*Hallas-Potts, Dawson & Herrington, 2019*) cells). Quantification methods combine measurement of migration on the surface of the surrogate tissue with measurement of invasion through it (Fig. 3C) and enable the study of differences in the patterns of migration and invasion (*Al Ameri et al., 2019*; *Paradiso et al., 2019*; *Borghese et al., 2020*) and the interaction with healthy cells (*Coelho et al., 2018*; *Hallas-Potts, Dawson*

*& Herrington, 2019*). The results are however generally reported as covered area over time, which can be misleading when cells move in three dimensions. As such, a combination of scratch wound healing and transwell assays should generally be preferred. A more extensive use of cell aggregates as the starting point for these assays, rather than single cells, is also warranted, as evidence consistently points to these structures as the most likely to initiate metastasis *in vivo* (*Steitz et al., 2020*; *Osborn et al., 2022*; *Rakina et al., 2022*; *Penet et al., 2018*).

For larger, scaffold-based cultures, the dynamic of ECM colonisation can also be evaluated. For example, in a study by Paradiso and colleagues, cells seeded on top of a 3D collagen scaffold progressively migrated to the bottom, following a cell-line specific dynamic (*Paradiso et al., 2019*).

The quantification of the expression of genes and proteins commonly associated with invasion and migration (*e.g.*, metalloproteinases) often accompanies the assays described above to study the molecular basis of migration and invasion. qPCR, Western blotting and immunohistochemistry, described in the next section, are the most commonly used techniques and can help to match underlying biochemical processes to the corresponding macroscopic behaviours.

Overall, the study of invasion and migration in a 3D setting is fairly mature, as techniques already developed for 2D cultures can be adapted to these new systems. Models comprising more than one cell type and complex ECM surrogates are the most interesting, as they yield the most transferrable results.

## GENE AND PROTEIN EXPRESSION

Gene and protein expression enables the study of the molecular basis of macroscopic phenomena and as such it is a fundamental tool of ovarian cancer research, from improving diagnosis (*Werner, Warton & Ford, 2022*; *Sheng et al., 2020*), to personalising treatment (*Hirst et al., 2018*; *Sapiezynski et al., 2016*; *Werner et al., 2024*) and studying drug resistance and disease progression (*Asante et al., 2020*; *Wang et al., 2019*).

Western blotting, quantitative PCR (q-PCR) and RNA sequencing (RNA seq) are the most widely used techniques, but protein staining, or the immunofluorescence study of relevant markers, are also an option (Table 3).

Overall, 3D cell culture has been shown to induce profound changes in gene expression when compared to monolayer settings. Markers connected with stemness, epithelial to mesenchymal transition and hypoxia response seem to be the most affected (*Al Ameri et al., 2019*; *Hirst et al., 2018*; *Song et al., 2020*; *Liu et al., 2018*; *Izar et al., 2020*), suggesting that a more complex culture setting might induce a more aggressive and resilient cell phenotype (*Zhang et al., 2021*).

The use of more *in vivo*- like models is advantageous for the study of gene expression, with primary cells maintained in 3D aggregates shown to preserve patient-specific expression patterns (*Velletri et al., 2022*). The results obtained using cell lines, however, are less conclusive with the expression patterns measured in 3D cultures differing from those of tissues of the same histotype (*Coelho et al., 2018*). Additionally, ECM composition, stiffness

**Table 3** Summary of the models and assays used to study gene and protein expression in 3D cell culture ovarian cancer models.

| 3D cell culture model | Assay | Citations |
|---|---|---|
| Cell aggregates | q-PCR | *Hirst et al. (2018)* |
| | RNA seq | *Singh et al. (2021)*, *Hirst et al. (2018)*, *Velletri et al. (2022)* |
| | Staining/Immunofluorescence | *Coelho et al. (2018)* |
| | Western blotting | *Singh et al. (2021)*, *Bahar et al. (2021)* |
| Spheroids | q-PCR | *AlAmeri et al. (2019)*; *Tofani et al. (2021)*, *Parashar et al. (2022)* |
| | RNA seq | *AlAmeri et al. (2019)*, *Izar et al. (2020)* |
| | Staining/Immunofluorescence | *Song et al. (2020)*, *Pietilä et al. (2021)*, *Heredia-Soto et al. (2018)*, *Hossein et al. (2019)* |
| | Western blotting | *Song et al. (2020)*, *Parashar et al. (2022)*, *Hossein et al. (2019)* |
| Scaffold | q-PCR | *Paradiso et al. (2019)*, *Liu et al. (2018)* |
| | Western blotting | *Liu et al. (2018)* |
| Organoid | microarray | *Hoffmann et al. (2020)* |
| | q-PCR | *Maenhoudt et al. (2020)*, *Hoffmann et al. (2020)*, *Lõhmussaar et al. (2020)* |
| | Western blotting | *Hoffmann et al. (2020)*, *Lõhmussaar et al. (2020)* |
| | RNA seq | *Kopper et al. (2019)*, *Hoffmann et al. (2020)*, *Lõhmussaar et al. (2020)*, *Zhang et al. (2021)* |
| | Staining/Immunofluorescence | *Hoffmann et al. (2020)*, *Lõhmussaar et al. (2020)* |

and fibrosis have been shown to influence gene expression in patient derived samples (*Pietilä et al., 2021*), but the same analysis in surrogate structures yielded less conclusive results (*Paradiso et al., 2019*; *Pietilä et al., 2021*). Finally, patient derived organoids have been shown, by multiple authors and with different techniques, to recapitulate the gene expression and mutation profiles measured in the tissue of origin (*Kopper et al., 2019*; *Maenhoudt et al., 2020*; *Hoffmann et al., 2020*; *Lõhmussaar et al., 2020*; *Zhang et al., 2021*).

Interaction between different cell populations may be a key element for achieving *in vivo*-like gene expression patterns. While few studies explore this aspect due to the technical difficulties in isolating the contribution of single cells, markers that were overexpressed in 3D cultures compared with 2D settings were shown to further increase when more than one cell type was present (*Velletri et al., 2022*). A more complex picture was painted by *Izar et al. (2020)* using single cell sequencing to characterise the gene expression profiles of the different cell types within ascites fluid (cancer, fibroblasts, macrophages, immune cells). While markers associated with cell cycle regulation, inflammation and immune response were shown to be commonly overexpressed, the key finding of this work was the high patient–patient variability which further highlights the importance of gene expression evaluation with sub-population resolution for treatment personalisation in OC.

Staining and immunofluorescence are generally regarded as more qualitative techniques, as highly trained personnel or specialised software are required to accurately quantitate the signal's intensity. They however provide valuable information regarding protein localisation within the cells and spatial variability within the culture, which cannot be obtained with the other techniques presented in this section. As changes in localisation result in functional

differences in protein activity, the combination of staining/immunofluorescence and a global evaluation of protein and RNA levels is expected to yield particularly relevant results.

The analysis of gene expression in 3D cell cultures benefits extensively from the technologies developed for 2D and *in vivo* models. This has enabled the widespread use of complex set-ups comprising multiple cell types, surrogate ECMs or patient-derived organoids. These have been shown to yield more accurate results when compared to simpler systems. While this further encourages the adoption of more realistic culturing set-ups, a systematic analysis of the differences between simpler models could be useful to identify which factors influence gene expression. Besides furthering our understanding of this complex process, the information could be used to devise innovative therapies that exploit regulatory factors and mechanisms already in place, to restore healthy gene expression patterns or accurately identify cancer cells.

## TREATMENT RESPONSE

Response to treatment encompasses all the assays and techniques that can be used to evaluate *in vitro* the effect of pharmacological agents (Table 4).

Cell aggregates are widely used to measure treatment response, as they couple simplicity of generation and use with the benefits of a 3D culture system. More complex set-ups, however, are necessary to investigate the role of different surrogate ECMs in drug response (*Hedegaard et al., 2020*; *Song et al., 2020*), the effect of treatment on macroscopic properties such as adhesion and invasion (*Lu et al., 2019*) or patient to patient variability (*Maenhoudt et al., 2020*) (Table 4).

Culturing cells in 3D is associated with an increased variability in cell behaviour and resistance to treatment (*Liu et al., 2020*; *Song et al., 2020*; *Tofani et al., 2020*; *Velletri et al., 2022*). This result is observed irrespectively of the experimental model, cell type and drug used, but is particularly evident in organoid models, which have been shown to recapitulate patient-specific features and drug response patterns (*Nanki et al., 2020*; *Maenhoudt et al., 2020*; *Hoffmann et al., 2020*; *Zhang et al., 2021*).

Non-uniform drug availability throughout the culture, caused by the finite diffusion velocity, is likely to be partially responsible for the differences observed in 3D. The coherence of the data measured in small cell aggregates with data obtained in larger structures, however, suggests that the environment itself might induce changes in cell behaviour, as the small aggregate size and lack of support structure in these models should pose little resistance to diffusion. Gene expression studies comparing cells cultured in 2D and 3D support this hypothesis (*Singh et al., 2021*; *Hirst et al., 2018*). Interestingly, exploiting these differences to identify new potential therapeutic targets has led to the reversion of the trend, showing cells cultured in 3D responding to targeted treatments better than their 2D counterparts (*Hirst et al., 2018*). Should this result be confirmed, it would offer a relatively simple strategy to improve therapeutic target identification in ovarian cancer and other diseases.

*Hirst et al. (2018)* offer another potential solution based on the lasting effects of maintaining cells in 3D. They show that pre-culturing cells in 3D leads to monolayer

**Table 4  Summary of the models and assays used to study response to treatment.**

| 3D cell culture model | Assay | Citations |
|---|---|---|
| Cell aggregate | Brightfield imaging, metabolic assay | *Pisano et al. (2020)* |
| | Cellular uptake, metabolic assay | *Michy et al. (2019)* |
| | Drug uptake, metabolic assay | *Van den Brand et al. (2020)* |
| | Ki67 staining, metabolic assay, q-PCR, RNA seq, colony forming assay | *Hirst et al. (2018)* |
| | Live cell monitoring, drug uptake, q-PCR, Western blotting | *Singh et al. (2021)* |
| | Live dead | *Brodeur et al. (2021)* |
| | Metabolic assay | *Velletri et al. (2022)* |
| | Metabolic assay, aggregate formation, mesothelial clearance | *Izar et al. (2020)* |
| | Metabolic activity caspase activity, cellular toxicity | *Hedemann N et al. (2021)* |
| | Metabolic assay, live dead, brightfield imaging | *Heredia-Soto et al. (2018)* |
| | Metabolic assay, spheroid volume, migration | *Borghese et al. (2020)* |
| Spheroids | Brightfield imaging, metabolic assay, Western blotting, mitochondrial membrane potential | *Bahar et al. (2021)* |
| | Live dead | *Song et al. (2020)* |
| | Metabolic assay | *Tofani et al. (2021)*, *Tofani et al. (2020)* |
| | Metabolic assay, spheroid size, cleaved caspase activity, WB, invasion and migration | *Hossein et al. (2019)* |
| | Metabolic assay, spheroid size and immunofluorescence staining | *Hedegaard et al. (2020)* |
| Scaffold-based | Metabolic assay, live dead | *Mohammad Hadi et al. (2020)* |
| Organotypic model | Adhesion, invasion and proliferation | *Kenny et al. (2020)* |
| | Adhesion, migration and invasion | *Ritch et al. (2022)* |
| | Cellular uptake, invasion, adhesion | *Lu et al. (2019)* |
| | Invasion and migration | *Joshi et al. (2021b)* |
| Organoid | Metabolic assay | *Nanki et al. (2020)*, *Cortesi & Giordano (2022)*, *Hoffmann et al. (2020)*, *Rai et al. (2018)*, *Nowacka et al. (2021)*, *Alday-Parejo et al. (2021)*, *Crivelli et al. (2017)* |

cultures with behaviour more closely approximating that obtained in cell aggregates. Pre-treating cells in this setting was also shown to further increase their *in vivo*-like behaviour.

Another important stream of research is drug delivery strategies aimed at reducing the systemic toxicity associated with traditional chemotherapy. 3D cell cultures have been instrumental in the development of these methods as they enable testing in simplified yet realistic settings. Among the proposed approaches, light-activation (*Van den Brand et al., 2020*; *Mohammad Hadi et al., 2020*) and nanoparticle carriers (*Singh et al., 2021*; *Lu et al., 2019*; *Michy et al., 2019*; *Joshi et al., 2021b*) are the most common. In both cases, the increased specificity toward cancer cells is expected to be particularly beneficial for the treatment of metastatic disease, where a heterogeneous mixture of healthy and cancerous cells needs to be targeted.

The high frequency of metastatic disease at ovarian cancer diagnosis has also promoted the use of co-culture models. These cultures enable the study the interaction of cancer

cells with healthy epithelium that occurs when new metastasis sites arise (*Tofani et al., 2021*; *Lu et al., 2019*; *Izar et al., 2020*; *Hossein et al., 2019*). Co-culture models are used to evaluate both how treatment affects the healthy cells and the effects of interaction between different cell types on treatment effectiveness. At the crossover of multi-cell-type models and innovative drug delivery strategies is the work presented by *Borghese et al. (2020)*, which relied on adipose-derived stem cells (ADSC), as vehicles for paclitaxel. These cells are able to incorporate and release the drug (*Melzer et al., 2019*; *Crivelli et al., 2017*; *Boyd et al., 2002*), with negligible alterations of their behaviour. This property, coupled with their ease of isolation and expansion, has made ADSC promising drug delivery systems. Indeed, in *Borghese et al. (2020)* paclitaxel released from ADSC was more effective than the same amount of paclitaxel dissolved in the media.

Drug testing and development is one of the major uses of 3D cultures, due to the immediate applicability of the benefits of improved accuracy. Consequently, it is expected to be associated with many of the innovations currently under development. Among them, methods with intra-culture resolution are likely to be particularly relevant, as they offer a way of investigating the higher variability observed when culturing cells in 3D. Mosaic spheroids could be a useful tool in this regard, being composed by a population of genetically heterogeneous cells (*Boyd et al., 2002*). The necessity of tagging each sub-population with a fluorescent marker restricts this analysis to cell lines, and could limit the number of genotypes that would be feasible to include. Nevertheless, they could provide valuable insights given how genetic heterogeneity is emerging as a main driver of ovarian cancer progression and drug resistance. More complex experimental models are also expected to become more common, as more comprehensive characterisations of the effect of different treatments will be conducted.

## CONCLUSIONS

In this review we have provided an overview of the main 3D cell culture models for the study of ovarian cancer *in vitro,* and we have shown that they span multiple levels of complexity to recapitulate different aspects of the disease. This wide variety of options is a great resource, as no single model, irrespective of its complexity, is going to perfectly replicate ovarian cancer. As such the combination of results obtained *via* multiple approaches has the potential to provide a more comprehensive picture of the investigated processes. A clear understanding of the assumptions and simplifications of each model is necessary however, to avoid misinterpretation of the results and aid the resolution of conflicting findings.

A key aspect still in need of development is the assays used to quantify the results. While important progress has been made in recent years both in terms of research prototypes (*Cortesi & Giordano, 2022*; *Dekkers et al., 2019*; *Wu et al., 2018*; *Schoppmeyer et al., 2018*; *Yao et al., 2022*; *Coluccio et al., 2019*; *Pan et al., 2019*) and commercial solutions (*e.g.,* IncuCyte), several methods still yield qualitative results only, while quantitative approaches still largely focus on population averages. Population averages can be misleading especially when they combine multiple, highly variable subpopulations. The current difficulty of monitoring cells over time, due to the almost ubiquitous requirement for cell fixation,

additionally contributes to the variability of results, as small differences in initial and environmental conditions cannot be accounted for. This limitation is however highly dependent on the specific experimental model, and is expected to be soon overcome, as devices capable of monitoring relevant cell properties over time, without affecting cell viability and behaviour, are being developed (*Lovecchio et al., 2019*; *Yao et al., 2022*; *Coluccio et al., 2019*; *Pan et al., 2019*).

Overall, 3D cell culture is revolutionising the *in vitro* study of ovarian cancer, providing simplified yet accurate models in which to study tumour cell function and test therapeutic strategies with results closely matching patient response. While work still needs to be done on many aspects of their development and use, 3D ovarian cancer models are expected to play a key role in the development of new therapies capable of improving patient outcome and thus reducing the burden of this terrible disease.

### Funding

This project received funding from the European Union's Horizon 2020 Research and Innovation Programme under the Marie Sklodowska-Curie grant assessment No 883172. The funders had no role in study design, data collection and analysis, decision to publish, or preparation of the manuscript.

### Grant Disclosures

The following grant information was disclosed by the authors:
European Union's Horizon 2020 Research and Innovation Programme: 883172.

### Competing Interests

The authors declare there are no competing interests.

### Author Contributions

- Marilisa Cortesi conceived and designed the experiments, performed the experiments, analyzed the data, prepared figures and/or tables, authored or reviewed drafts of the article, and approved the final draft.
- Kristina Warton conceived and designed the experiments, performed the experiments, prepared figures and/or tables, authored or reviewed drafts of the article, and approved the final draft.
- Caroline E. Ford conceived and designed the experiments, authored or reviewed drafts of the article, and approved the final draft.

### Data Availability

This is a literature review.

### Supplemental Information

Supplemental information for this article can be found online at http://dx.doi.org/10.7717/peerj.17603#supplemental-information.

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
