# Peer review of "Beyond 2D cell cultures: how 3D models are changing the in vitro study of ovarian cancer and how to make the most of them"

_PeerJ, doi:10.7717/peerj.17603_

## Round 0.1 · original submission · Major Revisions

Something of a split decision here - one reviewer is strongly in favour while the other is less enthusiastic. I am minded therefore to give you the opportunity to address the issues identified. Reviewer-2 comments are clear and should be straightforward. Reviewer 1 note that lack of context provided by no discussion of in vivo data (though I feel this does not need to be extensive, the point is well made), and clarity on the number of publication hits returned and their specificity to ovarian cancer.

I look forward to seeing the revised version in due course.

Reviewer 1 ·

Basic reporting

There is almost no background on the disease. There should also be a brief discussion of in vivo models since it would allow the author to discuss the merits of developing better in vitro models.

Experimental design

It is unclear how many hits were returned with the survey prompts. It will be beneficial to have some indications on whether or not 3D models for ovarian cancer or cancer in general are rising or not. This could provide some understanding of the current landscape for ovarian cancer research.

Validity of the findings

There should be more direct and focused discussion of the different models and assays in studying and modeling ovarian cancer.

Line 156 the term “simple 3D models” was not defined.

The author did not mention any specific methods of metabolic assays. Are 3D models able to recapitulate accurately? Do they show differences using 2D and 3D models?

Lines 183-191 require further elaboration on why this is significant.

Additional comments

There are multiple typos and inconsistencies.

·

Basic reporting

First two lines of abstract could be re-worded, point very valid but reads as jarring. Abstract clear and makes excellent points around importance of choosing correct methods to investigate cancer therapies. Introduction very well written, clear the points the authors are addressing and highlighting very important issues. Only suggestion is mention of hypoxia and complications caused as Ovarian cancer are highly hypoxia dependent and this often hinders the effects of therapies. Figures clear however it would read more easily figures were presented throughout text when introduced by authors. The figures add a great visual aid to readers and its important that readers not familiar with 3D techniques can easily visualizes what the authors are talking about. Paper well written throughout, with clear link to referenced techniques and methodologies as detailed in tables. Introduction supports the main subject of the paper, authors make clear the importance of this study and the impact that choosing the correct methods can have in the outcome of an investigation.

Experimental design

Study very relevant in field of cancer research, authors clearly state issues with methodologies and highlight the importance of maximizing impact of results by using the correct methods. Survey methodology not very detailed, authors could provide more information on what terms were used to search, only one example of culture set up searched. Authors could also state why work published within the last 5 years was included, was this to keep the paper relevant and up to date or was there a shift in the type of research being conducted, for example have more labs expanded on what models they are using post COVID. Following COVID there have been shortages and increases in price for consumable materials needed to create 3D model systems, for example; spinner flasks for spheroids growth have became very difficult to purchase as glass wear costs increase.

Validity of the findings

Authors address a pressing issue in cancer research and explain different aspects of 3D culture that can be assessed however there are a few additional points authors could mention to strengthen the paper;
In vivo models of ovarian cancer Section 82 - 166
Authors could mention the development of mosaic spheroids that can be used to assess different genetic alteration within a 3D mass, cells with different mutations can be used to form spheroids and tagged with fluorescent proteins to visualize (such as GFP) and to evaluate effects of treatment on heterogenetic model.
Authors could expand in this section on different methods of growing spheroids, lines 146-150 authors state 'These requirements preclude the possibility of longitudinal measurements and increase the
variability in the results, as differences in initial conditions (e.g., cell number and
distribution, material composition) cannot be accounted for'
This is not always true as there are many methods to form spheroids (spinner flask, 96 well round bottom plate, hang-drop method). 96 well round bottom plate is the most effective way to control the number of cells within a spheroid. For example 500 cells could be seed into each well and the spheroids left to form for the desired time - to generate required size. This method is also effective for monitoring the growth of spheroids and the change in volume over time. This is effective when comparing treated spheroids to untreated spheroids, the change in growth of the spheroids can be calculated and compared to the untreated control spheroids.
Section - Staining assays lines 221-242. Authors could mention the use of hypoxia or apoptosis (Caspase 3 detection) detection stains in immunohistochemistry (IHC). Hypoxia stain kits are available to purchase and required the treatment of spheroids for 2 hours with stain prior to fixation step in IHC. As such this can be used to investigate the desired diameter/volume needed to generate a hypoxic core within a spheroid. As such investigators are able to test each cell line and understand what size is needed and how long this takes to grow for respective cell lines. The effect of treatments on heterogeneous spheroids is then able to be investigated (proliferating cells., quiescent cells and hypoxic cells within one spheroids structure).
This would clash with point authors make on lines 250-253 (line 250 spelling error -Hirst), staining for a hypoxic core would correlate diameter to presence of hypoxia within spheroids.
Line 254- Discussion, could rename as end of section discussion as unexpected to have a discussion in middle of a paper.
Line 569-570- authors state that issues with 3D structures are that there is difficulty in monitoring the cells over time however the above points would solve this issue.

Additional comments

Overall the paper is very scientifically relevant and makes an important point, that choosing the correct method of investigation can have a huge impact on research. There are just a few points that the authors could include that would strengthen their argument and also add more detail regarding the potential that 3D models have.

---

## Round 0.2 · accepted · Accept

Thank you for carefully addressing the issues raised at review. I am satisfied with your responses which are clear and cogent. Hence, I recommend acceptance without further re-review.